# Low-Energy X-Ray Intraoperative Radiation Therapy (Lex-IORT) for Resected Brain Metastases: A Single-Institution Experience

**DOI:** 10.3390/cancers15010014

**Published:** 2022-12-20

**Authors:** Christian D. Diehl, Steffi U. Pigorsch, Jens Gempt, Sandro M. Krieg, Silvia Reitz, Maria Waltenberger, Melanie Barz, Hanno S. Meyer, Arthur Wagner, Jan Wilkens, Benedikt Wiestler, Claus Zimmer, Bernhard Meyer, Stephanie E. Combs

**Affiliations:** 1Department of Radiation Oncology, School of Medicine, Klinikum Rechts der Isar, Technical University of Munich (TUM), 81675 Munich, Germany; 2Institute of Radiation Medicine (IRM), Helmholtz Zentrum München, 85764 Neuherberg, Germany; 3Deutsches Konsortium für Translationale Krebsforschung (DKTK), DKTK Partner Site, 81675 Munich, Germany; 4Department of Neurosurgery, School of Medicine, Klinikum Rechts der Isar, Technical University of Munich (TUM), 81675 Munich, Germany; 5TUM-Neuroimaging Center, Klinikum Rechts der Isar, Technical University of Munich (TUM), 81675 Munich, Germany; 6Department of Diagnostic and Interventional Neuroradiology, School of Medicine, Klinikum Rechts der Isar, Technical University of Munich (TUM), 81675 Munich, Germany

**Keywords:** brain metastases, neurosurgical resection, intraoperative radiation therapy

## Abstract

**Simple Summary:**

Advances in systemic cancer management have improved survival for numerous types of solid cancer; therefore, the number of patients harboring BM is increasing. According to established guidelines, resection should be carried out in patients with single brain metastases and controlled primary disease or when histopathologic diagnosis is crucial for decision-making in cancer management. Post-surgery rates of local recurrence are high; hence, adjuvant local radiation therapy (RT) is indicated to improve outcomes. So far, there is no standard of care regarding dose and fractionation; furthermore, delineation of the cavity can be challenging. Lately, low-energy X-ray intraoperative radiation therapy (IORT) applied to the resection bed has emerged in clinical practice, offering local ablative treatment with steep dose gradients towards the surrounding healthy brain. We here retrospectively describe 18 patients with resected brain metastases, which had undergone IORT, demonstrating the effectiveness and safety of this technique in accordance with previous studies.

**Abstract:**

Background: Resection followed by local radiation therapy (RT) is the standard of care for symptomatic brain metastases. However, the optimal technique, fractionation scheme and dose are still being debated. Lately, low-energy X-ray intraoperative RT (lex-IORT) has been of increasing interest. Method: Eighteen consecutive patients undergoing BM resection followed by immediate lex-IORT with 16–30 Gy applied to the spherical applicator were retrospectively analyzed. Demographic, RT-specific, radiographic and clinical data were reviewed to evaluate the effectiveness and safety of IORT for BM. Descriptive statistics and Kaplan–Meyer analysis were applied. Results: The mean follow-up time was 10.8 months (range, 0–39 months). The estimated local control (LC), distant brain control (DBC) and overall survival (OS) at 12 months post IORT were 92.9% (95%-CI 79.3–100%), 71.4% (95%-CI 50.2–92.6%) and 58.0% (95%-CI 34.1–81.9%), respectively. Two patients developed radiation necrosis (11.1%) and wound infection (CTCAE grade III); both had additional adjuvant treatment after IORT. For five patients (27.8%), the time to the start or continuation of systemic treatment was ≤15 days and hence shorter than wound healing and adjuvant RT would have required. Conclusion: In accordance with previous series, this study demonstrates the effectiveness and safety of IORT in the management of brain metastases despite the small cohort and the retrospective characteristic of this analysis.

## 1. Introduction

In the management of cancer patients, brain metastases (BMs) are of increasing concern, secondary to better systemic treatment options, and hence, there is a higher risk of BM in a lifetime [1]. Based on epidemiologic data, 10–30% of all cancer patients will develop BM over the course of the disease [2,3,4]. When looking at specific entities, such as malignant melanoma or small cell lung cancer, incidences are up to 37% and 80%, respectively [5,6]. The most frequent primaries are small cell lung cancer or non-small cell lung cancer (NSCLC), breast cancer, malignant melanoma, renal cell carcinoma (RCC) and colorectal cancer, whereas the former three together account approximately for up to 80% [3,7]. BMs account for 50% of all intracranial neoplasia, and the incidence is ten-fold higher than that for gliomas [8]. Especially, small and asymptomatic BMs are amenable to stereotactic radiosurgery (SRS), but large or symptomatic lesions require microsurgical resection for enduring symptom relief [9,10]. It has been shown that maximal tumor resection improves survival [11]. However, the local recurrence rate after resection is up to 50%, secondary to remnant tumor cells [12,13]. Supra-marginal resection can even improve local control but has a higher risk of neurological deficits [14]. Early studies applying whole brain radiation therapy (WBRT) supported the idea of adjuvant RT to improve intracranial control but have the disadvantage of neurocognitive sequalae [12,15,16]. Local adjuvant stereotactic RT to the resection cavity has been demonstrated as an effective and safe treatment while preserving healthy brain tissue and hence neurocognition [17,18,19]. So far, several studies have demonstrated satisfying local control (LC) rates with postoperative local RT, which can be applied with single session radiosurgery or hypo-fractionated stereotactic RT (hFSRT) [20,21,22,23]. Nevertheless, there are some concerns about adjuvant RT: target delineation of resection cavities can be challenging due to the irregular shape, and the cavity size can vary in the early postoperative period [24,25,26]. Furthermore, the start of RT must be delayed for 1–3 weeks until adequate wound healing is completed; hence, systemic treatment needs to be delayed with a so far unknown impact on patient outcomes [26,27]. Therefore, low-energy X-ray intraoperative RT (lex-IORT) is gaining attention: surgical resection and RT performed within one procedure seems to be convenient regarding the duration of total treatment; at the same time, challenges of optimal target delineation for postsurgical cavities for external beam radiation therapy (EBRT) can be resolved [28]. Several studies have demonstrated effectiveness and safety with good local control rates up to 88% at 12 months [29,30,31]. This study is a retrospective descriptive single center analysis with a report of the first experience with lex-IORT for the management of BM at the Technical University of Munich (TUM) Medical Center in Munich, Germany (School of Medicine).

## 2. Materials and Methods

After obtaining the approval of the institutional ethical committee, 18 consecutive patients treated with lex-IORT after the resection of BMs between December 2017 and October 2020 were analyzed. The indication for surgical resection was set by the treating neurosurgeon approved by an interdisciplinary neuro-oncologic board when the BM was not amenable to SRS, with respect to size, perifocal edema, impending obstruction of cerebrospinal fluid flow and tumor location close to eloquent brain areas necessitating the long-term use of steroids for SRS. Patients with history of primary germ cell tumor, small cell carcinoma or lymphoma were excluded. Specific systemic treatment agents needed to be withheld for lex-IORT to reduce the risk of radiation necrosis (RN): BRAF/mitogen-activated protein kinase (MEK)-inhibitors ≥3 days, gemcitabine ≥7 days, trastuzumab-emtansine (T-DM1) ≥20 days, immune-checkpoint modulators (e.g., Nivolumab, Pembrolizumab, Atezolizumab) >1 day. Based on an intraoperative assessment, lex-IORT needed to be technically feasible and the histopathologic report based on frozen sections had to be compatible with BM, excluding a primary germ cell tumor, small cell carcinoma and lymphoma.

Immediately after tumor resection, all patients received lex-IORT with the INTRABEAM System PRS 500 with the low energy (50 kV) X-ray source XRS 4 (Carl Zeiss Meditec AG, Oberkochen, Germany) using intra-cavitary interchangeable spherical treatment applicators (diameter range 1.5–4.5 cm). The single-fraction treatment dose ranged from 16 to 30 Gy and was applied to the surface of the spherical applicator. Intraoperative navigation (Brainlab AG, Munich, Germany) and/or intraoperative magnetic resonance imaging (MRI) were applied to assess the shortest distance between the cavity edge and organs at risk (OARs), such as the brain stem and optical apparatus, to meet dose constraints according to Quantitative Analysis of Normal Tissue Effects in the Clinic (QUANTEC) [32].

Further, non-resected BMs were treated with SRS via volumetric modulated arc RT (VMAT) or via Gamma Knife SRS or with WBRT. Three patients received a lower IORT-dose with additional adjuvant hFSRT using VMAT plans.

Follow-up was generally performed with clinical examinations and a cranial MRI at 6 weeks and then every 3 months. MRI sequences applied were T1-native, 3D-T1-contrast enhanced, T2, T2 fluid attenuated inversion recovery (FLAIR) and if available diffusion weight imaging (DWI) and dynamic susceptibility contrast (DSC).

For assessments of local control, images were assessed based on an adaption of the response assessment criteria proposed by the Response Assessment in Neuro-Oncology Brain Metastases (RANO-BM) group [33]. The radiographical response was measured multidimensionally and was recorded for every resection cavity postoperatively at every follow-up visit. Lesions not treated with IORT were also assessed and followed up. Lesions were considered measurable when nodular contrast enhancement was visible on two or more axial slices and measurable with a minimum size of 5 mm (longest diameter (LD)). For measurable lesions >5 mm and <10 mm, an increase or decrease in the LD of <3 mm was considered stable disease (SD), and an increase of ≥3 mm compared to that at early post-operative MRI (within 48 h after resection) was considered progressive disease (PD). Regarding measurable lesions
≥10 mm, a minimum increase of 20% in the LD was considered PD; otherwise, it was considered SD. Distant brain control (DBC) was defined as the absence of new or progressive lesions at other sites independent of the IORT- site.

In cases of radiologic PD but a clinical suspicion of pseudo-progression, additional imaging, including a repeat exam, as well as advanced imaging technics (perfusion MRI, 18-fluor-ethyl-positron-emission tomography (FET-PET) and ultimately surgical pathology via biopsy or resection, were planned.

For assessments of safety parameters, the International Common Terminology Criteria for Adverse Events (CTCAE) version 5.0 for toxicity and adverse event reporting was used.

Medical records were reviewed for demographic (sex, age, grade-prognostic assessment (GPA) [1]), pathologic, oncologic treatment, RT planning and clinical outcome data. Dosimetric data were extracted from clinically approved and administered treatment plans using the treatment planning system (TPS) (Aria, version 13.0; Varian Medical Systems, Palo Alto, CA, USA, dose calculation AAA 13, dose grid spacing 2.5 mm) and Zeiss INTRABEAM© system PRS 500. 

All statistical analyses and the generation of graphs were performed using SPSS (version 24.0; IBM Inc., Armonk, NY, USA). Descriptive statistics were used to evaluate baseline characteristics including patient- and tumor-related characteristics, as well as doses and volumes investigated in the present study. Survival and local progression were estimated using the Kaplan–Meyer function; for local progression, patients dying without evidence of local recurrence and patients remaining free of local recurrence at the end of follow-up were censored.

## 3. Results

In this study, 18 total patients with 18 histopathologically proven brain metastases were analyzed. The female–to-male ratio was 10:8, mean age at IORT was 56 years (range, 19–72 years) and mean and median preoperative graduated prognostic assessment (GPA) score were 2.25 (range, 0.5–4) and 2.5, respectively. The most frequent primaries were NSCLC (*n* = 6) and malignant melanoma (*n* = 4). Tumor location was predominantly in the parietal (*n* = 6) and frontal (*n* = 5) lobe, respectively. Gross total resection (GTR) was achieved in all cases. Patient and tumor characteristics are summarized in Table 1.

The mean dose prescribed to the applicator surface was 21.3 Gy (range, 16–30 Gy). Applicators from 1.5 to 4 cm in diameter were used, in 13 out of 18 cases, size was 2–2.5 cm. The duration of treatment was a mean of 15 min and 33 s (range, 5:32–22:41 min). Three patients (no 16–18) received both 16 Gy intraoperatively and additionally adjuvant hFSRT 8 × 3 Gy (VMAT, 2 arcs, treatment percentage 95% isodose, normalization (N) D_99.8_ ≥ 95%), 10 × 3 Gy (VMAT, 1 arc, 100% IL, N D_99.0%_ = 100%) and 6 × 5 Gy (VMAT, 2 arcs, 95%, D_99.75_ ≥ 95%). Five patients received WBRT postoperatively due to number of BMs being ≥ 7 (*n* = 3) or postoperative clinical deterioration (*n* = 2). None of the patients had received cranial RT before IORT. RT characteristics are displayed in Table 2.

The mean follow-up time was 10.8 months (range, 0–39 months). The mean OS was 22.8 months (range, 0–39 months, 95%-CI 14.4–31.2 months), with two patients still alive at time of analysis (Figure 1).

There was one case with the recurrence of suspected local metastases (patient no 8) with linear contrast enhancement at 5 months post-IORT, which was diagnosed according to BM-RANO. There was no conformation with repeat MRI or advanced imaging due to extracranial progression and best supportive care. One case of leptomeningeal failure was observed (5.5%).

According to Kaplan–Mayer analysis, the estimated OS, LC and DBC at 12 months after IORT were 58.0% (95%-CI 34.1–81.9%), 92.9% (95%-CI 79.3–100%) and 71.4% (95%-CI 50.2–92.6%), respectively (Figure 1, Figure 2 and Figure 3). Excluding patients with additional RT (WBRT or hFSRT) to the cavity (*n* = 8), the estimated LC was 90.0% (95%-CI 71.4–100%) (Appendix A).

Two of three patients (No. 17 and 18) receiving adjuvant hFSRT to the cavity, in addition to IORT, who developed radiation and local wound infection, of CTCAE grade III, 12 and 5 months, respectively, after treatment had to undergo revision-surgery. One patient presented with a small cerebrospinal fluid (CSF) wound fistula of CTCAE grade II on the second postoperative day which was sutured with local anesthesia without further wound healing impairment over the course of the disease. In total, there were two patients with RN (11.1%).

One patient (No. 2) with BM of malignant melanoma developed a hemorrhage of two non-IORT lesions and died on the 20th postoperative day. Two patients (No. 6 and 18) were diagnosed with asymptomatic sinus venous thrombosis distant to the craniotomy on the first follow-up MRI. There were no other grade II or higher events related to IORT. Twelve patients received systemic treatment after surgery; the time between IORT and the initiation or continuation of systemic treatment was a mean of 53.5 days (range, 10–132 days).

## 4. Discussion

This is a report of the first experience with intraoperative RT (IORT) for 18 consecutively treated patients with resected brain metastases at our institution. In this series, we observed a mean overall survival of 22.8 months (range, 0–39 months, 95%-CI 14.4–31.2 months) and an estimated OS at 12 months of 58.0% (95%-CI 34.1–81.9%) (Figure 1). Applying the adapted RANO-BM criteria, local recurrence was observed in one case, generally translating into an estimated LC at 12 months of 92.9% (95%-CI 79.3–100%) (Figure 2). Five patients were treated with additional WBRT after IORT secondary to a high number of BMs or clinical deterioration. Three patients received lower IORT doses (16 Gy), but were treated with adjuvant local hypo-fractionated stereotactic RT (hFSRT) to the cavity. Therefore, looking at patients undergoing only IORT for RT to the resection bed, the estimated LC at 12 months is 90.0% (95%-CI 71.4–100%) (Appendix A). Six patients developed further BMs distant to the resection cavity; hence, the estimated DBC at 12 months was 71.4% (95%-CI 50.2–92.6%) (Figure 3). In this patient cohort, death was a strong competing event; seven patients died within the first 12 months mainly due to extracranial tumor progression. Since there was no dedicated exclusion for IORT, we also treated patients with low GPA mostly due to KPS and the extracranial tumor burden. Two of three patients receiving additional adjuvant hFSRT developed local infection and progressive RN and needed revision surgery. Both patients were predisposed to wound healing impairment: patient 17 was treated with the vascular endothelial growth factor inhibitor Axitinib, starting 10 days after IORT, and patient 18 had diabetes and was a heavy smoker (40 pack years). One patient had a CSF fistula, which could be treated locally. Overall, in our cohort, IORT was safe with no further CTCAE grade ≥2 events. There were no further cases of RN translating into an overall rate of radiation necrosis of 11.1% despite additional adjuvant WBRT in five cases. Nevertheless, due to the apparently higher risk for both wound healing impairment and RN, IORT and additional hFSRT should be used with caution.

There are a few published studies regarding Lex-IORT after the resection of BM; one prospective single-center study analyzed 23 patients with solitary resected BM receiving 14 Gy Lex-IORT to a depth of 2 mm. Seven patients had local failure at the cavity 9 ± 5.7 months after the operation, and three of them had the initial infiltration of the pia mater and tumor recurrence presented as leptomeningeal disease (LMD), and thus, the authors concluded that Lex-IORT was non-optimal in those cases. Three patients had histopathologically proven radiation necrosis, but all of those lesions had been previously irradiated with SRS or WBRT [31]. Cifarelli et al. published an analysis of retrospective multi-institutional Lex-IORT data from the US and Europe; 54 patients undergoing resection for large symptomatic BM (40% NSCLC, 38% frontal lobe) had received low-energy IORT with a median dose of 30 Gy to the applicator surface. LC, DBC and OS at 12 months were 88%, 58% and 73%, respectively with two (3%) patients developing LMD. The extent of resection was the only significant predictor for LC (GTR 94% vs. STR 62%), and the RN rate was 7% (four patients) [29]. Kahl et al. presented the retrospective data of 40 patients with 44 resected BMs treated with Lex-IORT; the median dose to the applicator surface was 20 Gy. The estimated 1-year LC, DBC and OS were 84.3%, 33.5% and 61.6%, respectively. RN was observed in one patient and four patients developed LMD over the course the disease [30]. Based on these studies, Lex-IORT seems to be effective and safe regarding LC and RN rates, comparable to historical hFSRT data.

Our described series has a smaller number of patients receiving IORT after the resection of BM; therefore, a dedicated statistical analysis was difficult and limited to descriptive statistics. OS in our cohort was lower than that in Cifarelli et al.’s pooled analysis, with an estimated overall survival at 12 months of 58% compared to 73%, and no advanced demographic data relating to tumor staging of the performance score were reported [29]. We had no specific patient selection criteria regarding performance scores, extracranial tumor burden and overall number of metastases. Half of the patients had ≥2 metastases and 13 of 18 patients had an active extracranial tumor, both translating into a lower GPA. Based on a large multi-institutional analysis of outcomes after postoperative FSRT, a controlled primary tumor is a prognostic factor significantly associated with survival [20]. Thus, these are the most likely reasons for impaired OS in this case series.

However, two of three patients having received additional hFSRT to the cavity presented with RN and wound infection over the course of the disease. We think that those concepts must be regarded critically, because of the high cumulative dose to the normal brain, bone and scalp.

The best practice in the radio-oncologic management of resected brain metastases is still under debate. There is no doubt about the need for additional RT to improve local control. Local postoperative RT to the resection cavity is showing good local control while sparing the healthy brain and therefore preserving neuro-cognition [17,18,19]. So far, several studies have demonstrated satisfying LC rates with postoperative local RT, which can be applied with single-session radiosurgery or hFSRT [20,21,22,23,34,35,36,37,38]. To date, there are no randomized controlled data available that could support either one of the two techniques being superior based on efficacy (i.e., LC) and safety (i.e., radiation necrosis (RN)). A recent multicenter analysis of 558 cavities treated with hFSRT (median total dose 30 Gy, median dose per fraction 6 Gy) demonstrated a 1-year LC rate of 84% and RN rate of 4.1% [20]. In general, adjuvant hFSRT seems favorable over single-fraction radiosurgery in terms of a higher LC and lower incidences of RN, most likely due to breaks between fractions and a higher biologic effective dose (BED) [20,39,40,41,42]. However, contouring of the surgical bed is challenging for hFSRT: despite the increasing use of local adjuvant RT, there is no definite standard for target delineation, and the cavity volume can vary over the time after surgery. Several studies have demonstrated variable changes in the cavity volume within weeks after surgery [24,25,43].

While focusing on accelerated local management of brain metastases with fewer in-hospital days and faster direction to adjuvant systemic treatment, IORT is brought into focus. Surgical resection and RT performed within one procedure seems to be convenient regarding the duration of total treatment; at the same time, challenges of optimal target delineation for postsurgical cavities for external beam radiation therapy (EBRT) can be negotiated. There are mainly two different techniques for local intra-cavitary RT for the management of brain metastases: first, brachytherapy with the application of permanent radio-isotope seeds (mainly iodine-125, infrequent caesium-131) into the cavity and closing the situs thereafter has excellent local control rates of 85–96% at 12 months after surgery but with high RN rates (up to 25%) [44,45,46,47,48]. Secondly, the lex-IORT device can deliver a high radiation dose via intra-cavitary X-ray applicators right after tumor resection. Lex-IORT seems to combine some potential advantageous characteristics: (i) elevated linear energy transfer (LET; compared to megavoltage X-rays) producing a higher rate of lethal DNA damage, such as double-strand breaks, and showing a higher relative biological effectiveness (RBE) [49,50,51]; (ii) a steep dose gradient with effective sparing of normal brain and further organs at risk (OARs); (iii) elimination of the interval between surgery and local RT with avoidance of potential cell repopulation and cavity remodeling, which are adverse effects on the LC in adjuvant stereotactic RT [24,52,53,54]; (iv) enhanced patient convenience secondary to shorter overall BM treatment durations and faster direction to systemic treatment. This case series supports those potential advantages of IORT to treat brain metastases with satisfying local control rates comparable to those with adjuvant treatment. In practice the IORT time was a mean of 15:33 min, and the surgery time was extended by 30–45 min; however, complications with anesthesiology or perioperative patient care were not observed. In five cases (27.8%), the time from surgery to the initiation or continuation of systemic treatment was ≤15 days, which is shorter than the regular overall time for adequate wound healing (10–14 days postOP) and hFSRT, which takes approximately 7–14 days, including planning. Nevertheless, these data must be seen with caution due to the low number of patients, the retrospective characteristic of the study and the heterogenous patient characteristics regarding the number of brain metastases and performance score. Additional postoperative hFSRT to the cavity seems to be very critical secondary to the high risk of radiation necrosis. Prospective randomized data are needed for a further evaluation of the best technique for perioperative RT and to determine which patients might benefit most from each treatment mode.

## 5. Conclusions

Despite the relatively small number of cases and the both descriptive and retrospective characteristics of this analysis, this study provides further evidence supporting the effectiveness and safety of lex-IORT in the management of resectable brain metastases. Therefore, a randomized trial comparing lex-IORT to radiotherapy of the resection cavity is currently in preparation.

## Figures and Tables

**Figure 1 cancers-15-00014-f001:**
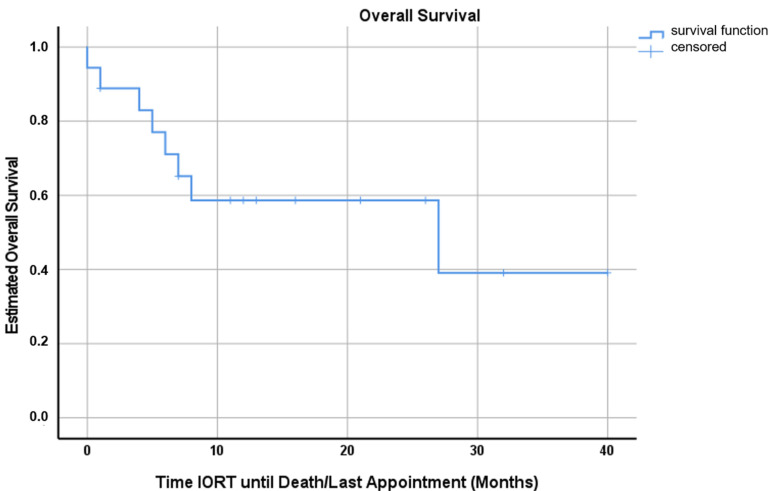
Kaplan–Meyer curve for overall survival with a mean OS of 22.8 months (range, 0–39 months, 95%-CI 14.4–31.2 months). At the time of analysis, two patients were still alive and on follow-up. The estimated OS at twelve months was 58.0% (95%-CI 34.1–81.9%).

**Figure 2 cancers-15-00014-f002:**
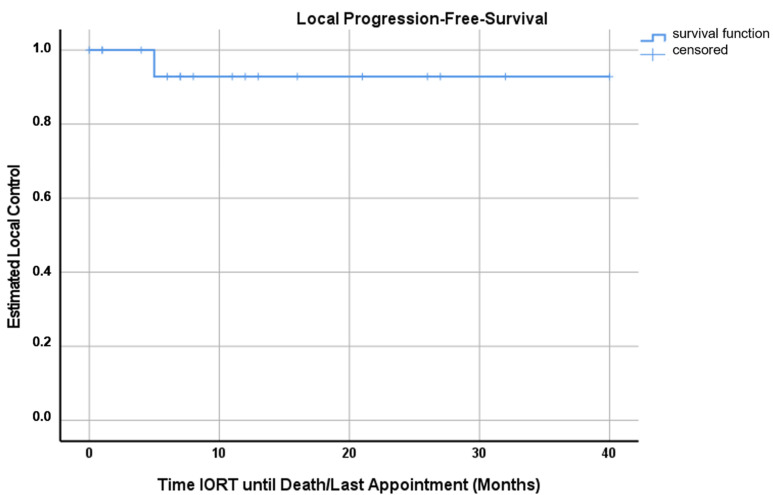
Kaplan–Meyer curve for local progression-free survival censoring local recurrence. For all 18 patients reviewed in this analysis, the estimated local control was 92.9% (95%-CI 79.3–100%).

**Figure 3 cancers-15-00014-f003:**
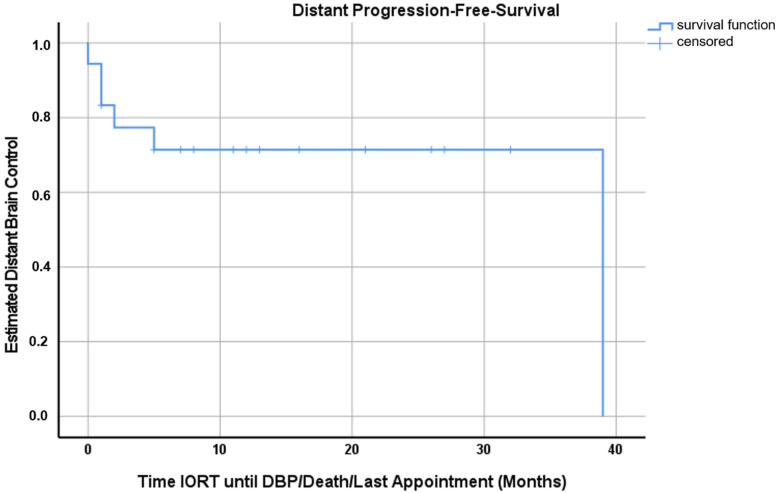
Kaplan–Meyer curve for distant progression-free survival. At one year after IORT, distant brain control was 71.4% (95%-CI 50.2–92.6%).

**Table 1 cancers-15-00014-t001:** Patient demographics and tumor characteristics for 18 patients undergoing IORT. TNBC: triple negative breast cancer, L: left, R: right front: frontal, temp: temporal, par: parietal, occ: occipital.

	Sex	Age	Primary	KPS	No of BMs	Extracranial Metastases	GPA	Location	Hemisphere	Largest Diameter (cm)			Dural Attachment	GTR
										ax	cor	sag		
1	F	37	TNBC	90	8	yes	2	occ	R	5.1	4.7	5.5	yes	yes
2	M	67	Melanoma	80	2	yes	1	occ	R	5.0	4.6	4.7	no	yes
3	F	55	NSCLC	90	7	yes	1	par	L	3.7	3.3	4.3	yes	yes
4	F	58	BC	70	15	yes	0.5	front	R	2.8	2.3	2.6	yes	yes
5	M	19	Osteosarcoma	80	1	yes	2.5	par	L	2.5	2.6	2.7	yes	yes
6	F	72	NSCLC	70	1	no	2.5	par	L	1.3	1.2	1.1	no	yes
7	F	74	RCC	80	1	yes	1.5	front	L	2.9	2.4	2.5	no	yes
8	F	51	Pancreatic Cancer	100	1	yes	3	par	R	2.5	2.4	2.4	yes	yes
9	F	54	NSCLC	100	1	no	4	par	R	3.7	3.3	3.8	no	yes
10	F	43	Melanoma	90	1	yes	3	occ	L	3	2.8	3.3	yes	yes
11	M	72	NSCLC	70	4	yes	0.5	front	L	2.6	2.2	2.5	no	yes
12	F	68	NSCLC	70	2	yes	1	par	L	3.3	2.9	3.6	yes	yes
13	M	55	Melanoma	100	1	no	4	temp	R	4.5	3.5	4.1	yes	yes
14	M	55	Urothelial Cancer	90	1	yes	4	front	L	2.5	3.1	3.1	no	yes
15	F	45	Melanoma	50	5	no	2	front	L	1.3	1.7	1.6	yes	yes
16	M	71	Rectal Cancer	90	2	no	2.5	temp	L	2.9	2.3	2.0	no	yes
17	M	55	RCC	100	1	yes	3	par	L	2.4	2.9	2.7	yes	yes
18	M	58	NSCLC	80	8	yes	2.5	front	L	2.6	2.9	2.6	yes	yes

**Table 2 cancers-15-00014-t002:** Radiation therapy related data for IORT and adjuvant RT and second-course RT.

Patient	Applicator Size (cm)	Dose (Gy)	Time (min:s)	postOP RTx			Second Course RTx
				WBRT	SRS	hFSRT (Cavity)	
1	4	20	24:10	10 × 3 Gy	-	-	SRS (2x)
2	3.5	20	17:59	-	-	-	-
3	2.5	20	18:10	10 × 3 Gy	-	-	WBRT
4	2	20	12:00	10 × 3 Gy	-	-	-
5	2	20	11:28	-	-	-	-
6	1.5	25	07:14	-	20 Gy	-	-
7	2.5	25	22:14	-	-	-	SRS
8	2	20	12:11	-	-	-	-
9	2.5	25	22:19	-	-	-	-
10	2	20	11:55	-	-	-	SRS
11	2	25	14:14	-	20 Gy (3x)	-	-
12	2.5	25	22:41	10 × 3 Gy	-	-	-
13	2.5	20	08:17	-	-	-	-
14	1.5	30	10:41	-	-	-	-
15	1.5	20	05:32	10 × 3 Gy	20 Gy (2x)	-	-
16	2.5	16	18:02	-	20 Gy	8 × 3 Gy	SRS (2x)
17	2	16	09:35	-	-	10 × 3 Gy	-
18	2	16	09:17	-	-	6 × 5 Gy	-

## Data Availability

The data presented in this study are available on reasonable request from the corresponding author.

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
