# Peer review of "Low-Energy X-Ray Intraoperative Radiation Therapy (Lex-IORT) for Resected Brain Metastases: A Single-Institution Experience"

_cancers, 2022, doi:10.3390/cancers15010014_

Round 1
Reviewer 1 Report
Dear authors,
Your article is eloquent and methodic, the topic chosen is interesting and fully debated. Please, find below some considerations:
- Is there a reason why your OS was lower than the other reported works?
- Can you better explain and motivate the following sentences: "Based on demographic data, extra-cranial and intracranial tumor burden, there was no specific selection for IORT, it is possible that the low GPA translated into a worse survival. Two patients received additional hFSRT to the cavity, both presented with RN and wound infection over the course of the disease. Hence those concepts must be regarded critically "?
- Please, enhance your point of view and what you want to add with your case series.
- This study is based on a small sample size and a larger one is mandatory to better understand the role of this procedure, please try to better discuss your results.
Author Response
Dear Editor,
Dear Reviewers,
We do appreciate immensely, that our manuscript “Low-energy X-ray intraoperative radiation therapy (lex-IORT) for resected brain metastases: A single-institution experience.“ (cancers- 2070929) has been considered for re-submission. Thank you for your helpful comments. We have put great efforts into the revision of our manuscript to thoroughly address the reviewers’ comments and questions to full satisfaction.
Thank you again for your consideration. We hope to hear back from you soon.
Yours sincerely,
Christian Diehl
Initial comment:
Your article is eloquent and methodic, the topic chosen is interesting and fully debated. Please, find below some considerations:
Answer to Initial Comment:
We would like to thank the reviewer for the overall positive evaluation of our manuscript.
Comment #1:
- Is there a reason why your OS was lower than the other reported works?
Answer to Comment #1:
We agree with your point on the minor overall survival compared to studies on IORT mentiones in the discussion section. Kahl et al reported on an estimated OS at 12 months of 62%, in our case series 1-year OS is 58%, we think the difference is minor[1]. Indeed, compared to Cifarelli et al. we did worse with 58% compared to 73%[2]. This study did not report on performance status, primary tumor control or overall number of brain metastases, all of which translating into the GPA and hence prognosis so we did hard to compare these data sufficiently. In our small series half of the patients had ³2 brain metastases and in 13 of 18 patients the primary was not controlled. According to a large multi-instituional analysis on hFSRT after resection of brain metastases a controlled primary tumor is a prognostic significantly associated with overall survival[3]. Furthermore, one patient died within 20 days after surgery with most likely impact on OS in this small cohort. These may be the main reasons for the inferior OS.
Comment #2:
- Can you better explain and motivate the following sentences: "Based on demographic data, extra-cranial and intracranial tumor burden, there was no specific selection for IORT, it is possible that the low GPA translated into a worse survival. Two patients received additional hFSRT to the cavity, both presented with RN and wound infection over the course of the disease. Hence those concepts must be regarded critically "?
- Please, enhance your point of view and what you want to add with your case series.
- This study is based on a small sample size and a larger one is mandatory to better understand the role of this procedure, please try to better discuss your results.
Answer to Comment #2:
We appreciate your comments, and we see the point about this sentence. We rephrased it and both re-arranged and modified the discussion section accordingly for better discussing our results and emphasizing our view. See lines 275-288 and lines 334 – 347. We hope these modifications address the reviewer´s thoughts appropriately.
References:
- Kahl KH, Balagiannis N, Hock M, Schill S, Roushan Z, Shiban E, Muller H, Grossert U, Konietzko I, Sommer B et al: Intraoperative radiotherapy with low-energy x-rays after neurosurgical resection of brain metastases-an Augsburg University Medical Center experience. Strahlenther Onkol 2021, 197(12):1124-1130.
- Cifarelli CP, Brehmer S, Vargo JA, Hack JD, Kahl KH, Sarria-Vargas G, Giordano FA: Intraoperative radiotherapy (IORT) for surgically resected brain metastases: outcome analysis of an international cooperative study. J Neurooncol 2019, 145(2):391-397.
- Eitz KA, Lo SS, Soliman H, Sahgal A, Theriault A, Pinkham MB, Foote MC, Song AJ, Shi W, Redmond KJ et al: Multi-institutional Analysis of Prognostic Factors and Outcomes After Hypofractionated Stereotactic Radiotherapy to the Resection Cavity in Patients With Brain Metastases. JAMA Oncol 2020, 6(12):1901-1909.
Reviewer 2 Report
This is a well written manuscript about a novel therapeutic approach. Intraoperative radiotherapy after the resection of brain metastases is a precise method, which can be perfomed with low side effects. All the necessary information has been given in clear tables and figures. Figure 4 could be omitted, since it does not add meaningful informations compared to Figure 1.
Author Response
Dear Editor,
Dear Reviewers,
We do appreciate immensely, that our manuscript “Low-energy X-ray intraoperative radiation therapy (lex-IORT) for resected brain metastases: A single-institution experience.“ (cancers- 2070929) has been considered for re-submission. Thank you for your helpful comments. We have put great efforts into the revision of our manuscript to thoroughly address the reviewers’ comments and questions to full satisfaction.
Thank you again for your consideration. We hope to hear back from you soon.
Yours sincerely,
Christian Diehl
Comments of Reviewer #2:
Comment #1:
This is a well written manuscript about a novel therapeutic approach. Intraoperative radiotherapy after the resection of brain metastases is a precise method, which can be perfomed with low side effects. All the necessary information has been given in clear tables and figures. Figure 4 could be omitted, since it does not add meaningful informations compared to Figure 1.
Answer to the Initial Comment:
We would like to thank the reviewer for the overall positive evaluation of our manuscript and agree with your point on figure 4. Instead of omitting it we added it as supplemental data at the end of the manuscript.
